# Structural Bases of Prion Variation in Yeast

**DOI:** 10.3390/ijms23105738

**Published:** 2022-05-20

**Authors:** Vitaly V. Kushnirov, Alexander A. Dergalev, Maya K. Alieva, Alexander I. Alexandrov

**Affiliations:** A.N. Bach Institute of Biochemistry, Federal Research Center “Fundamentals of Biotechnology” of the Russian Academy of Sciences, Moscow 119071, Russia; alexanderdergalioff@gmail.com (A.A.D.); atra.dingo@yandex.ru (M.K.A.); alexvir@gmail.com (A.I.A.)

**Keywords:** amyloid, prion, prion structure, yeast, Sup35, chaperones, Sup45

## Abstract

Amyloids are protein aggregates with a specific filamentous structure that are related to a number of human diseases, and also to some important physiological processes in animals and other kingdoms of life. Amyloids in yeast can stably propagate as heritable units, prions. Yeast prions are of interest both on their own and as a model for amyloids and prions in general. In this review, we consider the structure of yeast prions and its variation, how such structures determine the balance of aggregated and soluble prion protein through interaction with chaperones and how the aggregated state affects the non-prion functions of these proteins.

## 1. Amyloids and Prions—A General Description

Amyloids are filamentous protein aggregates, with each amyloid being comprised of a single type of protein. Amyloids grow by addition and structural conversion of soluble protein molecules. Amyloids have a regular cross-beta type structure and can be described as one-dimensional crystals. This term denotes structures that repeat in just one dimension, in contrast to common crystals, where the structure repeats in three dimensions.

Amyloid formation is regarded to be the cause of more than forty incurable age-related diseases, including Alzheimer’s and Parkinson’s diseases [1]. Due to their self-catalytic nature, some amyloids can be infectious, in which case they are termed prions, or proteinaceous infectious agents. Only one protein of humans and animals, PrP, is known to form a bona fide prion able to infect other organisms naturally [2]. However, many amyloids are able to spread within an organism, or certain tissues, thus showing prion-like properties (reviewed in [3,4]), and there is emerging evidence that some can also be infectious via direct injection [5,6,7,8].

Not all amyloids are pathological. Some serve useful functions (reviewed in [9]), like the CPEB amyloid of *Aplysia* or its *Drosophila* analog Orb2, which are involved in the formation of long-term memory [10,11].

The prion concept has been used to explain the nature of some yeast and fungal phenotypes inherited in a non-standard way [12]. In contrast to the mammalian prion, yeast prions are often quite neutral towards cells. They are even viewed by some authors as conditionally useful. Prions can be more or less detrimental to yeast under favorable environmental conditions, but they can increase the phenotypic diversity of a yeast population and thus allow appearance of cells better matching unfavorable conditions [13,14].

In contrast to globular proteins, which usually have just one stable fold [15], amyloids may have many distinct stably propagating folds, which manifest in different physico-chemical properties and different prion phenotypes, as observed for PrP [16], Aβ peptide [17], synuclein [18], and yeast Sup35 prion [19,20,21]. Such structurally and phenotypically different prions are referred to as «strains» in mammalian PrP studies and as «variants» in the case of yeast prions, to avoid confusion with strains of yeast. Thus, prions may be considered as protein-based “genes” with a number of “alleles”, which differ not in the sequence of amino acids, but rather in the manner in which the polypeptide is folded. In contrast to traditional nucleic acid-based genes, there is insufficient current understanding of the mechanisms by which such proteinaceous “genes” decode amyloid polymorphism into the variations of observable phenotypes. This review will consider the variation of yeast prions and possible mechanisms relating this variation to amyloid structure.

## 2. Types of Amyloid Architecture

A characteristic property of amyloid fibrils is cross-beta type intermolecular structure. In some amyloids this structure involves the whole protomer, while in others, only part of it is involved. Amyloids of the former group are usually made of small proteins or peptides, like Aβ, while the latter are typically made of larger polypeptides, like yeast prion proteins. In yeast prions, non-amyloid domains are likely to retain their structures and functions [22], though their activities are usually greatly reduced (Section 6.1). However, some proteins are functional in their amyloid state, rather than the soluble state, including the “memory protein” CPEB [23] (Orb2 in Drosophila [24]), probably because their function requires several functional domains acting in close proximity. The activity of the prion form may differ from both that of the normal protein and its absence, as in the case of the yeast protein Swi1 [25].

In a cross-beta structure, beta sheets are parallel to the fibril axis, while beta strands are perpendicular to it. This structure can be implemented in several types of arrangement: parallel-in-register, parallel-out-of-register, helical (β-solenoid) and anti-parallel.

A unique and underappreciated feature of all amyloids, except for the helical ones, is that each protomer in them comprises only one strand of intermolecular beta sheet and thus a protomer occupies only a tiny space of 0.5 nm along a fibril. For comparison, in actin filaments each F-actin molecule occupies about 5.5 nm along the filament [26], or 11 times more (Figure 1). Such dense packing of protomers in amyloids, and in yeast prions in particular, leaves a very narrow space for entities to interact with the amyloid and could pose restrictions on the interaction of amyloids with chaperones and other proteins [27] (see Section 5.5).

While the exact spatial structure has not yet been established for any yeast prion, one key feature of these structures is quite clear: the part of each protomer involved in the intermolecular beta sheets can be regarded as a one-strand-thick two-dimensional structure (which is not necessarily perfectly flat).

### 2.1. In-Register Arrangement

The parallel in-register type is probably the most frequent arrangement for amyloids, in which beta strands of adjacent protomers are located in parallel and the corresponding residues from adjacent protomers are close to each other (in register) [28,29]. Of note, being “in-register” unambiguously defines being “parallel”, so the latter word is redundant in this definition and we will omit it. Why is such an arrangement the most frequent? Any amyloid core fold should satisfy two conditions: it should represent a local energy minimum and reproduce the fold along a fibril. In terms of energy, the in-register arrangement is favorable for hydrophobic and aromatic residues, but is unfavorable for charged residues, which would experience repulsion when placed in close proximity. Charged residues are rare in yeast prions, but less so in human amyloids. For example, synuclein contains eleven charged residues per sixty-residue amyloid core, of which six are presumed to form the intermolecular beta sheet (PDB: 2N0A) [30]. Another consideration in favor of the frequent occurrence of the in-register structure could be that it is the simplest arrangement allowing precise sequential reproduction of the fold.

### 2.2. Helical, or Solenoid Type

In a solenoid amyloid structure a protomer occupies two or more layers of intermolecular beta-sheets. For example, in the [Het-s] prion of the filamentous fungus *Podospora anserina* a protomer makes two turns of a helix [31]. A similar structure was found in the HELLF, another prion of *P. anserina*, distantly related to Het-s [32]. Importantly, the two turns of the solenoid in each of these prions are formed by similar but not identical sequences. Thus, in solenoid amyloids, similar to in-register amyloids, the amyloid fold is transferred along a fibril between similar sequences. A mass-per length measurement for the extracellular curli (CsgA) amyloid of *E. coli* indicated one protomer per 1 nm, which suggests a solenoid structure with two beta strands per protomer [33]. Notably, CsgA contains five imperfect repeats, rather than two. Quite interestingly, artificial solenoid amyloids can be made from proteins with beta-helical structure, such as antifreeze proteins [34,35], simply by removing their end turns, which seem to restrict intermolecular bonding. Such amyloids can contain four or more helical turns per protomer. As in other cases, the turns in these structures are formed by imperfect oligopeptide repeats.

In contrast to these cases, a four-rung β-solenoid model had been regarded the most likely for the fibrils of mammalian PrP protein [36], despite the fact that the PrP prion core does not contain repeating sequences and would thus violate the similarity principle. It was eventually shown that brain-derived PrP prion has an in-register, rather than solenoid, structure [37].

Solenoid amyloids are rare compared to in-register ones. The in-register structure is typical of pathogenic amyloids and those lacking a clear function, including mammalian amyloids [38] and yeast prions, exemplified by Rnq1 [39], Ure2 [40], Sup35 [29] and “scrambled” proteins with computer-randomized Sup35 N domain sequences [41]. In contrast, the solenoid structure can be found among functional amyloids like [Het-s] or curli [31,33]. It is possible to suggest that the solenoid structure in these cases was naturally selected to better fit their function, which may favor increased spacing between protomers and their functional units.

No phenotypic or structural variants have been reported for the [Het-s] prion or other solenoid amyloids. Apparently, solenoid structures might be less capable of forming variants than in-register structures, due to the constraint of forming intramolecular bonds between the repeats in a certain pre-defined manner. An interesting consequence of this idea is that the presence of multiple prion variants, as in the case of PrP, suggests an in-register rather than a solenoid structure.

### 2.3. Other Types

The antiparallel arrangement is often found in amyloids formed by uniform sequences, like polyglutamine or polyalanine [42,43], and such an arrangement appears to be more thermodynamically stable that the parallel [42]. Islet amyloid polypeptide (IAPP) can form parallel in-register, out-of-register, and antiparallel amyloids [44,45]. An unusual Aβ amyloid structure was reported, in which residues 30–42 form antiparallel beta sheet, while residues 11–24 form parallel out-of-register beta sheet [46].

Polyglutamine amyloids are likely to have another interesting property. For these, the term “in-register” loses meaning, since in any register the sequence is the same. Due to this, it is reasonable to assume that protomers in polyglutamine amyloids do not observe register [47]. An important potential consequence of this would be that proteins with sufficiently long polyglutamine tracts could form branching amyloids, where any half of a polyglutamine region can independently seed amyloid growth. Importantly, such branching, atypical of other amyloids, can sometimes be observed [48]. The non-register structure of polyglutamine amyloids is also supported by their nearly linear, rather than sigmoidal, thermal melting curve [47]. The staggered non-register relative location of polyglutamine protomers should, in theory, lead to partial loss of structural information along the fibril and a lack of stably propagating structural variants. However, such variants have been described [49].

## 3. Yeast Prions, a Model for Prion/Amyloid Studies

Yeast prions have been a convenient and fruitful model for studying prions and amyloids. Though they cannot address many specific medical aspects of amyloidoses, yeast prions are well suited for the study of fundamental properties of amyloids and prions, their appearance, propagation, interaction with chaperones etc., and offer a highly convenient, rapid, cheap and safe alternative to animal models. In addition, yeast can be used to study prion properties of known and presumed amyloids from various kingdoms of life.

The first two yeast prions, [*PSI*+] and [*URE3*], were discovered in 1965 by Brian Cox and 1971 by Francois Lacroute, respectively, as phenotypes with unusual non-Mendelian genetic behavior [50,51], and their prion nature was postulated by Reed Wickner only in 1994 [12]. Further experiments confirmed that [*PSI*+] and [*URE3*] are amyloid-based prions of translation termination factor eRF3 (Sup35) and nitrogen metabolism regulator Ure2, respectively [52,53,54].

The [*PSI*+] phenotype demonstrates increased nonsense codon readthrough due to transition of Sup35 from a soluble state to poorly functional prion aggregates. This phenotype can be conveniently detected in *ade1–14* or *ade2–1* nonsense mutants, which normally do not grow in the absence of adenine, and produce red colonies due to accumulation of a red intermediate of adenine biosynthesis. [*PSI*+] *ade* cells can grow in the absence of adenine and yield white or pink colonies. The latter phenotype reveals the existence of prion variants and allows them to be distinguished by the extent of the translation termination defect.

Like most other yeast prion proteins, Sup35 has a modular structure with separate prionogenic and functional domains. Its N-terminal (N) domain of residues 1–123 defines the Sup35 prion properties, while the functional C-terminal (C) domain (254–685) performs the essential function in translation termination. The middle (M) domain (124–253) is not critical for either the Sup35 function or its prion properties [53,55]. Such modular structure, combined with convenient phenotypic detection, allows the testing of prionogenic properties of other proteins. This is performed by replacing the Sup35 N domain with the subject proteins or their fragments. The characteristic property of most yeast prion domains (PrD), including Sup35, is a high content of glutamine and asparagine (QN). The prion properties of a hundred yeast QN-rich domains were tested by fusing them to the Sup35C domain, and 19 fusions showed such properties [56]. The prion properties of the CPEB protein of *Aplysia californica*, involved in the animal’s long-term memory, were tested in a similar way [57]. This protein behaves as a functional amyloid in *Aplysia*, but in yeast acts as a bona fide prion giving phenotypically different variants. The Sup35 system also revealed the prion behavior of some mammalian [58], plant [59], archaeal [60] and baculovirus [61] proteins.

[*URE3*] prion allows the utilization of a poor nitrogen source, ureidosuccinic acid (USA), in the presence of a rich source, ammonia. The Ure2 protein blocks USA utilization by binding the transcriptional activator Gln3 involved in nitrogen metabolism, and inactivating it by sequestering it in the cytoplasm [12,62]. The prion form of Ure2 cannot efficiently bind Gln3, which allows growth on USA-containing medium. [*URE3*] has variants, as described in Section 7.1. Ure2 has an N-terminal PrD (1–71) and the C-terminal functional domain (90–354) showing similarity to glutathione transferase.

The yeast [*PIN*+] prion (Psi INduction) was discovered to be a factor strongly required for the appearance of [*PSI*+] [63]. Although it was later found that most prions can facilitate the appearance of other prions, the Rnq1 prion appears to be the most efficient in seeding [*PSI*+] [64,65]. [*PIN*+] therefore usually denotes the Rnq1 prion, so in this paper we will use the more specific and strict [*RNQ*+] designation. Thus, the [*RNQ*+] prion has a “gain-of-function” phenotype, which is so far the only known function of Rnq1. The [*RNQ*+] prion can be helpful to yeast by allowing the appearance of other prions and thus provide better adaptation to various adverse environmental conditions [14]. [*RNQ*+] variants exist, differing in their efficiency of [*PSI*+] induction [66]. In contrast to Sup35 and Ure2, the Rnq1 PrD is located in the C-terminal part of the protein (residues 153–405).

Along with [*PSI*+], [*URE*3] and [*RNQ*+], several other amyloid-based prions have been described in *S. cerevisiae*, such as [*SWI*+] [67], [*OCT*+] [68], [*MOD*+] [69], [*LSB*+] [70], [*MOT3*+] [56] and [*NUP100*+] [71]. However, these prions lack easily detectable phenotypes, and so very little is known about their variants.

To complete the picture, the [*GAR*+] prion [72] and nearly 50 other prions [73] that relate to uncharacterized but presumably non-amyloid structural conversion of proteins have been discovered in yeast. These prions can significantly increase the phenotypic variation of yeast, but due to their unknown nature, they will not be considered here.

## 4. [*PSI*+] Variants

### 4.1. Discrimination of [PSI+] Variants

Variation of [*PSI*+] has been studied far better than that of any other yeast prion. Initially, and for a long time after their discovery, the [*PSI*+] variants were distinguished solely by the strength of their nonsense suppressor phenotype [74], and were divided into two main categories, “weak” and “strong”. Weak [*PSI*+] prions allow *ade* mutants to grow slowly on medium lacking adenine and produce pink colonies. Strong [*PSI*+] prions exhibit faster growth on Ade- medium and white colony color.

In practice, it is difficult to discriminate more than three [*PSI*+] variants based only on suppression. One reason for this is that suppression can be affected by genetic and epigenetic factors. For example, a noticeable proportion of [*PSI*+] isolates carry duplication of chromosome 1 with the *ade1–14* gene, which doubles the apparent suppression level [75]. The [*RNQ*+] and [*SWI*+] prions both reduce production of eRF1(Sup45), the key factor of translation termination, and thus they should strengthen the [*PSI*+] phenotype. When present together, these prions demonstrate significant suppression even in a [*psi*-] background, which was initially described as the [*NSI*+] prion [76]. Notably, [*RNQ*+] is usually present in [*PSI*+] cells, while the presence of [*SWI*+] in these is usually not monitored.

Some of the weak and strong [*PSI*+] isolates may be indistinguishable by colony color [20]. However, we developed three tests to distinguish these two classes unambiguously. Firstly, there is a reproducible difference in the pattern of proteinase K digestion. Secondly, multiple copies of the *SUP35* gene are lethal for strong [*PSI*+], but not detrimental for weak [*PSI*+]. Thirdly, multiple copies of the *HSP104* gene readily cure weak [*PSI*+], but not strong [*PSI*+] [20].

Prion variants can also be distinguished by their sensitivity to altered levels of other chaperones [77,78,79]. This method and other described approaches allows the differences between certain variants to be established. A more complicated task is to ascribe a certain identity to prion variants, to establish then similar identities of [*PSI*+] isolates obtained independently and possibly from different backgrounds. This task was addressed by King and coauthors, who offered a reliable way to distinguish and identify wide range of [*PSI*+] variants based on interaction of the prion Sup35 with a set of Sup35 N domain mutants and GFP fusions [21,80]. This approach allowed up to 23 distinct and stable [*PSI*+] variants to be distinguished. However, the authors note that only five of these variants can be obtained in the “standard way”, i.e., through Sup35 overproduction. The remaining variants were obtained by passing these [*PSI*+] variants from wild-type Sup35 to Sup35 mutants and back [21].

### 4.2. [PSI+] Variants Obtained in Modified Backgrounds

However, the number of amyloid folds that Sup35 can acquire is even higher. Because a prion state decreases the function of Sup35, which is an essential protein, it is reasonable to expect that some [*PSI*+] variants should be lethal to cells. Such variants were observed by McGlinchey et al. in the presence of low-copy rescue plasmid encoding Sup35C protein lacking the N and M domains under control of a repressible promoter. Lacking its prion domain, this protein was seen to provide a constant source of soluble and functional Sup35 activity. Cells with lethal [*PSI*+] were unable to lose the plasmid. Among newly obtained [*PSI*+] isolates, such lethal [*PSI*+] constituted about 8%, while about 46% more of the [*PSI*+] isolates grew poorly in the absence of Sup35C [81]. Regrettably, the cause of this lethality and whether it is solely related to the Sup35 prion fold, rather than any additional factors, remains unclear.

A series of studies by the Wickner lab and coauthors showed that the absence or reduced activity of several proteins allow the appearance of [*PSI*+] variants that are unable to propagate in the wild-type background. These proteins are ribosome-associated Hsp70 Ssb1 and Ssb2 [82], Hsp104 weakened by T160M mutation [83], Upf1,2,3 nonsense-mediated mRNA decay factors [84] and Siw14 pyrophosphatase [85]. For the [*URE3*] prion, nearly all isolates appearing in the *∆btn2∆cur1* double mutant were unable to propagate in the wild-type background [86,87]. Btn2 and its paralog Cur1 promote prion curing by enhancing aggregation of prion particles. These works allowed the authors to conclude that there are many other [*PSI*^+^] variants, whose propagation is restricted by wild-type expression of the mentioned proteins, which thus represent anti-prion systems blocking detrimental [*PSI*+] variants (reviewed in [88]).

However, at least in one case, the conclusion may be formulated in a different sense. Huang et al. analyzed the [*PSI*+] variants obtained in the presence of the weakened Hsp104 T160M mutant, mentioned above [83]. Four [*PSI*+] variants appeared in this background, and three of them were novel and unable to propagate in the presence of wild-type Hsp104 [89]. The fourth variant was a standard strong VH variant, which showed weaker suppression in the Hsp104 T160M background. The three new variants were unstable in the presence of wild-type Hsp104 and converted into standard VH, VK and VL [*PSI*+] variants, as described earlier [80]. In contrast, these VH, VK and VL could not propagate in the mutant background due to insufficient Hsp104 activity. Thus, this case is best interpreted by assuming that every genetic background favors a certain subset of Sup35 prion folds and disfavors others, rather than in terms of an anti-prion activity of Hsp104. In line with such an interpretation are the observations that Hsp104, Sis1 and Arg82 [85,90,91] are required for propagation of all or almost all [*PSI*+] variants, but are never regarded as “pro-prion” systems. Curiously, *ARG82* is the nearest neighbor of *SUP35* on chromosome 4. A further observation is that strong [*PSI*+] becomes lethal in the absence of the *PUB1* and *UPF1* genes. In this case the mutant rather than the wild-type background restricts strong [*PSI*+] [92].

The frequency of prion appearance is usually several-fold higher in the mentioned strains with weakened or absent anti-prion proteins [83,88]. This indicates in favor of the anti-prion systems approach, and so possibly both concepts may be correct, depending on the particular “anti-prion” protein.

In addition to prion variants propagated in altered backgrounds, some Sup35 amyloid folds exist that are able to propagate in vitro, but cannot stably propagate in vivo (described in Section 5.3.4).

## 5. Sup35 Variant Prion Structures

### 5.1. Methodological Note

The Sup35 prion structure was probed with various physical and biochemical methods: fluorescent labeling [93], Hydrogen/Deuterium exchange combined with solution nuclear magnetic resonance (NMR) [94], solid state NMR [95,96] and proteinase K (PK) digestion with subsequent mass-spectrometric identification of the PK-resistant peptides [20,97]. A weak side of these important studies is the source of studied Sup35 amyloids: most of them used self-seeded fibrils of bacterially produced Sup35NM generated in vitro, based on observation that the fibrils assembled at 4 °C or 25–37 °C induce strong or weak [*PSI*+], respectively, when introduced to yeast [98]. However, the conformational uniformity of such fibrils is difficult to guarantee. A better approach, used by Frederick et al. [95], is to seed recombinant Sup35 with Sup35 prion isolated from yeast, but even then the folds of such fibrils may noticeably differ from the prototype prion [99]. Therefore, it is preferable to use Sup35 prions isolated from yeast, and we used this approach to map the PK-resistant structures of Sup35 prions [20].

PK mapping is a relatively simple and affordable approach, with some important advantages. PK is a 29 kDa protein, and so it can relatively well model prion accessibility to chaperones, which are of comparable, though somewhat larger, size. It allows estimation of which parts of a protomer are structured and which are unfolded and so can attract chaperones, which is the key information defining their interaction with the prion that eventually defines phenotype. PK is a precise instrument and contrary to some published data can cut after any residue except for proline, though it does show moderate residue preferences [20].

It may appear that the ultimate structural solution would be structure reconstruction based on cryo-EM or NMR. However, it is possible that some structures are insufficiently rigid or their orientation is not firmly fixed, allowing them to escape recognition. For example, the cryo-EM structure of Orb2 amyloid resolves only a 30-residue region of the 162-residue prion-like domain [100].

Further discussion of Sup35 prion structure is largely based on our PK mapping [20], which is the only work to have studied many variants of this structure.

### 5.2. Overall Prion Structure of the N and M Domains

Analysis of the Sup35 prion structures resistant to PK in 26 prion isolates obtained in different ways revealed that such structures can appear within four regions, residues 2–72, 73–124, 125–153, and 154–221 (Cores 1 to 4) [20]. Importantly, only the N-terminal of these structures (Core 1) is present in all [*PSI*+] isolates. A Core 2 structure at residues 91–121 is more frequent in weak [*PSI*+], while Core 3 at residues 124–147 is more frequent in strong [*PSI*+]. All four structures together were observed only once in the W8 [*PSI*+] variant. The PK data for Cores 1 and 2 are largely in agreement with earlier studies using fluorescent labels [93] and Hydrogen/Deuterium exchange [94], though with minor and explainable discrepancies.

The omnipresent N-terminal Core 1 is also the key determinant of the [*PSI*+] phenotype. We observed two types of Core 1 readily distinguishable by the PK digestion pattern, corresponding respectively to strong and weak [*PSI*+] isolates. Thus, the presence of distal Cores 2 to 4 has little effect on the phenotype, and this appears strange. It is natural to assume that unfolded regions within the prion domain should serve as a target for chaperones and thus increase the frequency of fragmentation. This idea is confirmed by the observation that deletion of some oligopeptide repeats from the Sup35 N domain, which tend to be unfolded, decreased fragmentation, increased size of Sup35 prions and turned a strong [*PSI*+] into phenotypically weak [47,101]. Why then does the [*PSI*+] phenotype not depend on the presence or absence of Core 2 and Core 3 structures? One possible answer is that such structures are present in all [*PSI*^+^] variants, but they could be soft and relatively PK-sensitive, and this possibility requires exploration. Another possibility is that the unfolded regions distant from the N terminus are less accessible to chaperones due to fibril architecture.

Some [*PSI*+] variants appear to be heterogeneous for Cores 2 to 4. This manifests as the simultaneous presence of Core 2 at overlapping locations, e.g., 82–100 and 91–119. Also, low levels of the peptides corresponding to Cores 2, 3 or 4 in many [*PSI*+] isolates could indicate their presence only in a part of the prion population. This agrees with some electron microscopic observations of the structural heterogeneity of Sup35 fibrils seeded in vitro by [*PSI*+] lysates [102], and might complicate future detailed structural studies.

### 5.3. Core 1 Structure

The Core 1 (residues 2–72) can be divided in two parts, 1A (2–32) and 1B (33–72). The Core 1A is resistant to PK, while Core 1B is partially sensitive. Furthermore, almost no internal fragments of Core 1B were observed in PK digests, which indicates that this part is rapidly degraded by PK after being cleaved by PK from the Core 1A [20]. What kind of structure is the Core 1B?

#### 5.3.1. Amyloid β-Sheet Regions Can Be Smaller Than Those Protected from PK

To understand how to interpret the partial protection from PK, it is worth studying the case of the [Het-s] prion. Its prion-forming region of residues 218–289 is fully protected from PK [103]. However, according to the established spatial structure of this region, the intermolecular β-solenoid of Het-s is formed by two 21-residue imperfect repeats, while the remaining 30 residues are unstructured, including the large loop between the repeats [31]. Thus, either the loop in the structure is more structured than has been presumed, or the loop is protected by somehow sticking to the β-solenoid core. Data comparison for synuclein shows that the PK-protected regions are larger [104,105] than the intermolecular β-sheet in the established structures [30,106].

The described data allow the suggestion that intermolecular cross-beta structure is restricted to Core1A, while Core 1B does not form such structure and is protected from PK only by virtue of its association with Core 1A. This roughly agrees with the excimer fluorescence data for Sup35NM indicating that the region of residues 43–85 does not form intermolecular in-register structure [93].

#### 5.3.2. Proline Residues Could Restrict Amyloid β-Sheet Structures

It should be noted that Core 1B is mainly formed by two-and-a-half oligopeptide repeats with the consensus sequence PQGGYQ(Q)QYN. The remaining repeats (region 73–96) are either PK-sensitive or form a lateral part of Core 2 that is probably analogous to Core 1B. This suggests that the repeats have a low propensity to form amyloid beta-sheet structures and thus mainly serve to mediate chaperone recognition and fragmentation. This agrees with earlier observations that deletion of some repeats reduces fragmentation and softens the suppressor phenotype without altering the prion fold [101].

The low amyloidogenicity of the repeats could be related to the presence in each of proline residues poorly compatible with beta structure. Of note, proline is absent in the β-solenoid forming repeats of Het-s and its 22 homologs from different species [107], in analogous repeats of HELLF [32], in the repeats of β-solenoid antifreeze proteins [34], in the amyloid beta, and synuclein amyloid core [30]. Among the Sup35 cores, proline is absent from Core 1A, the central part of Core 2 (residues 95–118) and from a larger part of Core 3 (125–137), though several prolines are present in Core 4.

#### 5.3.3. Probable Nested Structure of the Core 1 Variants

The difference between the Core 1 PK digestion patterns, which can distinguish strong and weak [*PSI*+] variants, though fully reproducible, was not very dramatic. These digests contain the same peptides, but in different proportions [20]. The difference may be roughly represented as the presence in weak [*PSI*+] prion folds of major PK-sensitive sites after residues 42 and/or 45. This can explain the two most evident distinctions of the weak structures: the dominance of peptides 2–42 and 2–45 over 2–34 and 2–37, and the significantly lower amount of the 2–70, 2–71 and 2–72 peptides. This allows suggestion that strong and weak [*PSI*+] variants share a similar key element in their Core 1 structures, but differ in some additional elements or details.

#### 5.3.4. On the Importance of the Core 1

Core 1 is critically important for [*PSI*+] propagation and defines the phenotype. None of the 26 studied [*PSI*+] isolates lacked the Core 1, while deletion of the first 30 Sup35 residues reduced about fourfold the proportion of Sup35 in amyloid form upon high overproduction in the [*RNQ*+] background. This amyloid was based solely on Core 2 [20]. In contrast, in vitro Sup35 does not show such restriction. Ohhashi et al. described two Sup35 amyloid folds with cores at residues 81–148 or 62–144, lacking the protease-resistant N-terminal Core 1. These folds were initially obtained using Sup35NM with a S17R anti-prion mutation, but they were able to propagate stably on wild-type Sup35NM. However, when introduced into yeast, these amyloids produced unstable prions [97] that rapidly convert into a standard strong [*PSI*+] [21], which, according to our work, has an N-terminal core, but lacks a large middle core [20] and so completely differs from the starting amyloid fold.

Sup35 can propagate as a prion in mammalian cells, but strikingly, this does not require the Sup35 N-terminal region, and depends on the region corresponding to the Core 2 [108]. The key difference between the yeast and mammalian chaperones is the powerful fragmenting chaperone Hsp104, and so it appears to be a crucial indicator defining the importance of Core 1 in yeast. It may be, for example, that Core 1 in any possible fold is too robust to be fragmented in the absence of Hsp104.

Finally, we observed that the [*PSI*+] phenotype is defined mainly, if not fully by the amyloid structure of the N-terminal region 2–72. The mutations used to discriminate between a large number of distinct [*PSI*+] variants lie within the same region [21]. However, we observed that [*PSI*+] variants can also differ in their distal prion structures, which are phenotypically nearly silent. These distal structures are inherited stably in some [*PSI*+] variants, though may be unstable in some other variants [20]. Thus, distal cores introduce a large, though silent, increase in the variety of Sup35 prion folds, and it is debatable whether or not this additional variation should be regarded as a strain difference.

The terminal location of prionogenic regions is favored in yeast prions and is important for their properties. Of the nine bona fide yeast prions, in five (Sup35, Ure2, Swi1, Mot3 and Nup100) this region is located at the N-terminus, in two (Rnq1 and Lsb2) at the C-terminus, and in just two (Cyc8 and Mod5) is internal. Furthermore, placing of the Sup35 Core 1 region inside the protein by fusing glutathione transferase to the N terminus completely blocks the Sup35 prion properties [109]. The exposure of the Core2 region at the C-terminus due to nonsense mutations increases the rate of spontaneous [*PSI*+] appearance about 6000-fold [110]. The internal QN-rich fragment of Gln3 can form a prion when placed at the N terminus, while full Gln3 cannot do this [56,111]. Thus, the terminal location of a prionogenic region strongly increases its prion potential in yeast, though it is not fully clear why. Possibly, terminal location increases conformational flexibility, thus favoring easier sampling of different potentially amyloid folds.

### 5.4. Contradictory Structures of the M Domain

It has been generally assumed that the formation of Sup35 prion structures is restricted to the QN-rich N domain, while the M domain, rich in charged residues, is incapable of forming in-register intermolecular interactions, and so never forms amyloids on its own, either in vivo or in vitro. While our discovery of Core 3 and Core 4 structures in the M domain contradicts this, some earlier observations from other laboratories have also suggested the possibility of amyloid structures in the M domain, when the N domain adopts a prion structure. NMR analysis of Sup35 fibrils generated in vitro labeled with ^13^C-Leu indicated that the majority if not all seven Leu residues of the M domain are located in the in-register β-sheet [29]. Electron microscopic observation of Sup35NM fibrils seeded in vitro by [*PSI*+] yeast lysates revealed the presence of “thick” and “thin” fibrils. These data suggest that the M domain is unfolded in thin fibrils, but largely folded in thick fibrils [102]. The folded state of the M domain is faithfully propagated along thick fibrils, and the amyloid fold seems the best if not the only way of propagating a certain structure along a fibril. Otherwise, one should expect a mixture of folded and unfolded M domains within a fibril.

Thus, the observation of large PK- and Sarcosyl-resistant Core 3 and 4 structures in the M domain [20] is not very surprising, though proof is required of their amyloid nature. Formation of such structures should make a Sup35 fibril more compact, like the observed thick fibrils [102], which should leave much less space for Sup35 interactions with chaperones, Sup45 and the ribosome (Figure 1). This should affect prion fragmentation and disassembly, thus affecting the Sup35 turnover between the soluble and aggregated states, levels of soluble Sup35 and functional Sup45, and the functional activity of Sup35 aggregates. However, the contradictory point is that no such effects were observed.

### 5.5. Structural Reconstruction of the Sup35 Prion Particle

Currently there is sufficient information to reconstruct the actual proportions of the Sup35 prion particle (fibril) and the proteins interacting with it (Figure 1), and this can provide some novel insights about its properties. A key feature of in-register fibrils and the Sup35 prion in particular is that every protomer occupies a tiny space of just 0.47 nm along a fibril. Such dense packing of protomers leaves a very narrow space for the molecules interacting with the amyloid, and can pose significant restrictions on these interactions. The size of the Sup35C domain (48kDa) is about 6 nm, and its essential partner Sup45 (49 kDa) is of similar size. The sizes of chaperones involved in prion fragmentation, Ssa (Hsp70, 70 kDa), Sis1 (Hsp40, 2 × 38 kDa) dimer and the Hsp104 hexamer (6 × 102 kDa) are even larger.

The Ssa1 and Ssa2 proteins are present in ex vivo Sup35 fibrils in a proportion of one Ssa molecule per two Sup35 [112], and the same value has been observed for synuclein amyloid [27]. The likely target of Ssa binding is the unfolded region between Cores 1 and 2. However, the reconstruction in Figure 1 suggests that there is insufficient space for binding of that amount of Ssa to this region, and so possibly Ssa also targets the M domain.

Fragmentation of the Sup35 prion relies on sequential action of chaperones Sis1, Ssa and Hsp104 [113]. It is known that overproduction of Hsp104 does not improve fragmentation of Sup35 [114]. The reason for this may be that it is impossible to accommodate more Ssa1/2 or Hsp104 complexes around a prion fibril.

**Figure 1 ijms-23-05738-f001:**
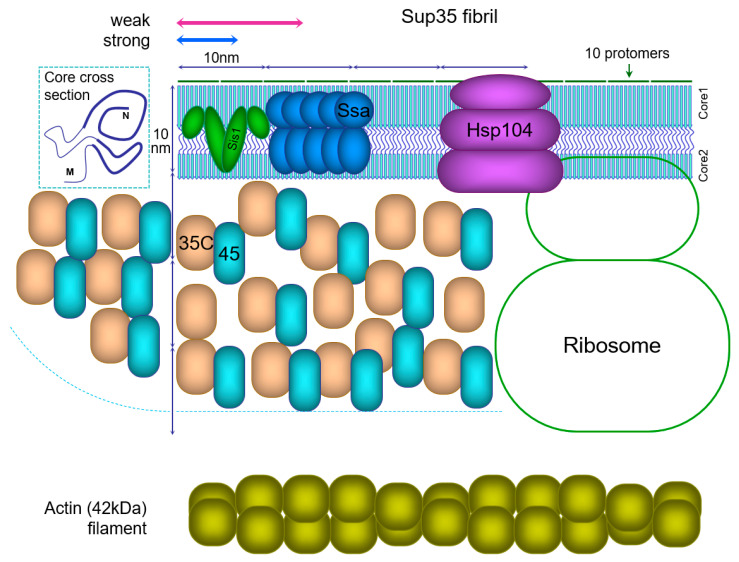
Schematic structure of the Sup35 prion fibril and associated proteins. All sizes are drawn to scale. A possible cross-section of the fibril core is presented in the top left corner, with PK-resistant cores shown by thick lines. For the Sup35 fibril the PK-resistant Cores 1 and 2 are shown joined by an unfolded region, but the M domain is not shown. The actual number of Sup35C (35C) domains together with Sup45 (45) protein should be 4-fold higher than shown. Sup45 is bound to many but not all Sup35C domains. A long Sup35 fibril is shown, but the approximate size of Sup35 prion fibrils is indicated above by arrows: 10–30 protomers for strong [*PSI*+] or 20–50 for weak [114]. Sis1 chaperone dimer is likely to be present at one per ten protomers, and the Ssa chaperones at one per two protomers [27,112], but only half of Ssa molecules are shown. Though a recent in vitro study on Sup35 fibrils suggested a ratio of one Sis1 dimer to two Ssa, these data were not directly quantified [113]. Hsp104 hexamer and the ribosome outline are shown. The actin filament is shown for comparison, to emphasize the tight packing of protomers in amyloids.

The requirement to accommodate the Sup35C domain may pose restrictions on structure formation in the M domain. “Thick” Sup35NM fibrils were reported, in which nearly the whole M domain appears to be folded [102]. However, this places the Sup35C domain at a too small radius around the fibril, where there is likely insufficient space to accommodate it. Thus, Sup35C would interfere with complete folding of the M domain. The observation of thick Sup35 fibrils, as well as the evidence for amyloid structure in the M domain [29] could be an artifact of using Sup35NM fibrils instead of full Sup35. In our PK mapping experiments [20] we used Sup35NM-GFP, which is a more correct instrument, though not perfect, because GFP (27 kDa) is smaller than Sup35C (48 kDa) or Sup35 bound to Sup45 (48 + 49 kDa).

Notably, Sup35 fibrils in this reconstruction and in reality [112] are so short that they do not actually resemble fibrils. Sup35 prion oligomers include on average about 15 protomers in strong [*PSI*+] and 30 in weak variants [114]. This makes 7 and 14 nm long fibrils, comparable to the amyloid core diameter of about 10 nm [102,115]. The Sup35C domain can occupy the space around the fibril with 60–65 nm diameter [102,115]. This space can be viewed as a cylinder and two side hemispheres with radius R of ~31 nm, and cylinder length L of 7 or 14 nm. L equals 0.47 nm ∗ N, where N is the number of protomers. The volume of the cylinder is πR^2^L, and the volume of two hemispheres is 4/3πR^3^. Volume available per protomer to accommodate Sup35C and Sup45 is V= πR^2^ (L + 4/3R)/N, or, substituting, πR^2^ (0.47 nm + 41 nm/N). According to this, the volume V decreases significantly with the number of protomers. For a short prion with N = 10 the volume V available for Sup35C is about tenfold higher at the expense of side hemispheres than such volume for a long fibril with N = ∞. This implies that in longer fibrils Sup35 is less accessible to Sup45 and chaperones. This could explain, at least in part, why long Sup35 fibrils found upon Sup35 overproduction are poorly fragmented in vivo [116], and, also in part, why strong Sup35 fibrils deplete Sup45 more efficiently [20] and are better fragmented than weak ones.

## 6. The Mechanisms Relating Sup35 Prion Structure to Phenotype

The [*PSI*+] phenotype, i.e., the translational readthrough, is defined by the proportion of Sup35 remaining soluble, the possible functionality of aggregated Sup35 and the ability of aggregated Sup35 to sequester the Sup45 (eRF1) protein.

### 6.1. The Balance of Soluble and Aggregated Sup35

In any cell harboring a [*PSI*+] variant, Sup35 exists in two fractions, soluble and prion. Soluble Sup35 can constitute about 5–15% of the total in weak [*PSI*+] isolates and 0.3–0.8% in strong ones [117]. Since Sup35 is an essential protein, it may appear that the latter values are incompatible with viability. However, yeast is surprisingly tolerant to low Sup35: a study of *SUP35* nonsense mutations revealed a mutant with only 0.5% of the wild-type Sup35 level [118]. Importantly, aggregated Sup35 retains some functionality [22], which should support viability of strong [*PSI*+] cells.

Notably, the method used for quantifying soluble Sup35 [117] is far from being perfect, since (a) a significant portion of Sup35 is removed from lysate prior to this assay as “debris” in both [*PSI*+] and [*psi*-] cases; (b) it is difficult to correctly separate prion and non-prion Sup35, since the latter can participate in large ribosomal and non-ribosomal complexes [119], making it difficult to establish the correct centrifugation stringency. A more correct test would be to measure the proportion of Sup35 able to enter an SDS-acrylamide gel without prior boiling of the sample [120], but such measurement has not been reported.

The proportion of soluble (non-prion) to prion Sup35 results from an interplay of two factors: the Sup35 variant-specific fibril growth rate and the number of fibrils. The growth rate depends on the inherent ability of amyloid fibrils to elongate (the rate constant), on the concentration of soluble Sup35, and the ability of chaperones to interfere with polymerization. The former parameter can be determined in vitro, while the latter has not been probed experimentally. Notably, there appears to be enormous discrepancy between experimental data. In weak variants Sup35 prion polymers are about twofold longer [114], so their number is halved. Considering the more than 20-fold lower level of soluble Sup35 in strong variants [117] and twofold higher number of fibril ends, the rate constant for fibril growth is tenfold higher in strong [*PSI*+]. However, this constant was measured in vitro for Sup35 fibrils formed at 4 °C regarded as equivalent to strong [*PSI*+] fibrils, and the growth was fivefold slower than growth of weak fibrils [121]. Thus, the discrepancy in the data was 10 × 5 = 50-fold. Most likely, the error arose from the in vitro experiment, being the most sophisticated and thus more error-prone, or due to crowding effects and/or chaperones having a considerable effect in vivo. A model was offered, relating the key parameters of prion polymerization: the levels of soluble and aggregated Sup35 depending on constants for fibril growth, fibril fragmentation and certain other parameters [121].

### 6.2. Causes for the Reduced Translation Termination Activity of the Sup35 Prion Form

The main [*PSI*+] phenotype is translational readthrough, or decreased translation termination. Two key players in the translation termination are Sup35(eRF3) and its partner Sup45(eRF1). Sup45 catalyzes termination, while Sup35 facilitates this process and provides energy through GTP hydrolysis. Both proteins are essential for termination and cell viability, though Sup45 appears more critical, in vitro at least [122]. The minimal observed life-compatible level of Sup45 is 8% compared to 0.5% for Sup35 [118,123].

The reduction of Sup35 activity in the prion form is likely to occur via two mechanisms. The prion structure is restricted to the Sup35N and possibly M domains. It is presumed, though has not been shown directly, that the structure of the Sup35C domain required for translation termination is not affected. However, a prion particle brings many Sup35 molecules into a confined space (Section 5.5, Figure 1) and so an equivalent number of ribosomes cannot be accommodated around it, and only a small fraction of Sup35 can act at any given moment. Steric interference of Sup35 functional C domains could also occur. Furthermore, Sup35 involved in large prion particles cannot diffuse through the cell efficiently in order to reach a terminating ribosome.

Importantly, Sup35 strongly interacts with its partner, Sup45, and can efficiently deplete Sup45 from the soluble fraction. This effect is illustrated by observations that a twofold increase in Sup35 level strongly impairs growth of strong [*PSI*+] variants, and a multiple increase is lethal, while a balanced increase of Sup35 and Sup45 is not toxic and can even improve translation termination [20,124,125]. Thus, one can assume that in cells with strong [*PSI*+] the activities of both Sup35 and Sup45 are inhibited simultaneously by the two mechanisms described above for Sup35. In contrast, weak [*PSI*+] variants seem to sequester Sup45 less efficiently, since multicopy *SUP35* (about 20 copies) does not cause sufficient Sup45 depletion to affect the cell growth.

While the translation termination activity may come from both soluble and aggregated Sup35, it appears that in strong [*PSI*+] isolates this activity comes mainly from aggregated Sup35 [22] due to the extremely low level of soluble Sup35. In weak [*PSI*+] variants, the level of soluble Sup35 is much higher, while the terminating activity of aggregates should be lower due to their larger size, and so it is likely that this activity comes mainly from soluble Sup35.

## 7. [*URE3*] and [*RNQ*+] Variants

### 7.1. [URE3] Prion Variants

[*URE3*], the prion form of the Ure2 protein, allows utilization of the poor nitrogen source USA in the presence of ammonia. However, this phenotype does not allow easy discrimination of [*URE3*] variants. To distinguish them, the *ADE2* gene was placed under the control of *DAL5* promoter. *DAL5* transcription is positively regulated by Gln3, which, in turn, is sequestered from the nucleus by binding to monomeric Ure2. This gives a red color to non-prion colonies, while [*URE3*] colonies are pink or white [126], which resembles the color indication of [*PSI*+] variants. This allowed discrimination of up to four [*URE3*] prion variants by colony color, and also provided easy means to evaluate the variant-specific rate of prion loss. The authors divided [*URE3*] variants into two types, A and B. The variants with white or pink colonies were classified as type A, while type B isolates were almost the same shade of red as prion-less cells.

Some interesting parallels can be noted between the [*URE3*] and the [*PSI*+] variants. Type B variants and weak [*PSI*+] have higher levels of soluble monomer and lower mitotic stability than type A [*URE3*] and strong [*PSI*+], respectively. This suggests that these parameters generally correlate among prions. Overproduction of full-length Ure2 yields both type A and type B isolates, whereas overproduction of Ure2 PrD generates only type B prions [126]. In a similar way, Sup35 PrD overproduction generated only 4% of strong [*PSI*+], while that of full Sup35 generated 60% [127].

The idea that the folding of the functional domain is not altered upon prion formation is best studied using Ure2. The Ure2 C-terminal functional domain shows a distinct homology to glutathione transferases. However, it demonstrates glutathione-dependent peroxidase activity, rather than transferase function. Upon Ure2 amyloid formation in vitro, the peroxidase activity does not change significantly [128]. Similar observations were made for barnase, carbonic anhydrase, glutathione S-transferase, and GFP fused to the Ure2 PrD. The activity of these proteins was at most mildly reduced in the amyloid form, indicating that they retained their native structures [129]. Thus, it can be stated that prion formation does not block functions related to small substrates, in contrast to interaction with large molecules and their complexes. The latter authors proposed that Ure2 interaction with Gln3 is sterically blocked by the prion structure of the Ure2 PrD. This differs somewhat from the quantitative mechanism of Sup35 inhibition in [*PSI*+] cells (Section 6.2).

### 7.2. Incorrect Use of PrD Hybrid Proteins

The fusions of various prion domains to Sup35(M)C or fluorescent proteins have been extensively used as reporters for studying prion properties of such domains. However useful, this approach should be used with caution, because it is likely to work poorly and give erroneous results when C-terminal prion domains, like in the case of Rnq1, are placed in N-terminal positions in reporter fusions.

This concern originates from observations that many yeast prion domains are asymmetric, similarly to Sup35, and a relatively small terminal region is more important for prion properties than the remaining QN-rich region. Only the first 37 residues (or ~10%) of Swi1 PrD are sufficient for prion aggregation in [*SWI*+] cells, while the lack of these residues abolishes aggregation [130]. The Rnq1 PK-resistant structure was a fortunate by-product of our study of Sup35 prion structures, and we invariably observed that in [*PSI*+] preparations, most of which were also [*RNQ*+], the Rnq1 C-terminal core of 40 residues (16% of PrD) was fully protected from PK, while internal Rnq1 fragments were seen only occasionally [20].

A further related observation suggests that placing such terminal sequences inside a protein inhibits their prion properties partially or completely, while shifting of internal QN-rich sequences to a terminal position enhances such properties (see Section 5.3.4) [109,110,111].

Regrettably, the majority of studies of Rnq1 prion variants rely on the use of Rnq1 PrD-Sup35MC or Rnq1 PrD-GFP hybrid proteins, allowing monitoring of the efficiency of prion polymerization as nonsense readthrough or aggregation (e.g., [66,131,132,133] and others). However, we argue that it is highly likely, if not certain, that such hybrid proteins manifest the prion properties of either the N-terminal or internal Rnq1 PrD regions, while the “native” core formation at the C terminus is inhibited by the Sup35 or GFP appendage and so the observed [*RNQ*+] prion is different from the “natural” one. A landmark survey of the prion properties of QN-rich yeast proteins also included testing of C-terminal and internal domains as N-terminal fusions with Sup35C [56]. Such results should be viewed with caution.

### 7.3. [RNQ+] Prion Variants

The primary phenotype for discriminating [*RNQ*+] variants is their efficiency in facilitating [*PSI*+] formation. According to this phenotype, several [*RNQ*+] variants were described, named as “low”, “high” and “very high” [66], as well as “very low” and “medium ” [131]. To observe whether these variants differed in the efficiency of prion polymerization, a hybrid reporter protein was constructed by replacing the Sup35 N domain with the Rnq1 PrD (153–405). Then, different levels of soluble reporter should manifest in different colony colors. Using this test, some correlation was observed, with less efficient [*RNQ*+] showing generally darker shades of pink. However, when endogenous Rnq1 was removed from such strains, the phenotypes changed and the correlation disappeared [133], which indicates that the reporter did not faithfully reproduce the original fold. Another phenotype is the pattern of aggregation of Rnq1 tagged with GFP. In some variants nearly all Rnq1 aggregates are sequestered into a single dot, located at the IPOD (Insoluble PrOtein Deposit [134]); in others Rnq1 forms multiple dots [135].

As we noted above, the reporter Rnq1 fusions to Sup35MC or GFP are unlikely to reproduce the original [*RNQ*+] fold. Therefore, the data described above can only be used as an indication of [*RNQ*+] variant differences, and do not allow any conclusions on the actual properties of these variants.

[*RNQ*+] variants also differ according to the spectrum of induced [*PSI*+] variants. For example, [*RNQ*+]-high preferentially seeds strong [*PSI*+] variants, while [*RNQ*+]-low is biased towards weak [*PSI*+] formation [136]. Other distinctive [*RNQ*+] phenotypes include mitotic stability, size, and thermal stability of Rnq1 amyloids. Overall at least 12 different [*RNQ*+] variants can be distinguished [137].

However, no dedicated study of the Rnq1 prion structures was conducted, and it is not known whether there are any prion cores in the Rnq1 PrD besides that of the C-terminal. It is also not clear which part of Rnq1 forms the interface involved in cross-seeding of Sup35 prion polymerization.

## 8. Conclusions

In this review, we have aimed to provide a close-up view of the yeast amyloid structure, particularly its possible variation and the mechanisms translating a prion structure into the phenotype. While the general picture is relatively clear, some important questions remain to be answered.

Why are the terminal regions of prion domains the most important for prion properties and the phenotype? Do the interactions of unfolded regions with chaperones depend on the distance from the protein terminus?Why is Core 1 of the Sup35 prion so critical in yeast, but is not required for, or can even harm, prion propagation in mammalian cells?Why do prionogenic regions show much stronger prion properties when located in terminal positions, rather than inside the protein sequence, and do they acquire different structures in these two cases?

These questions should and can be answered in the future.

## Data Availability

Not applicable.

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
