# Peer review of "Structural Bases of Prion Variation in Yeast"

_ijms, 2022, doi:10.3390/ijms23105738_

Round 1
Reviewer 1 Report
Review of “Structural Bases of Prion Variation in Yeast” by Vitaly Kushnirov et al.
The authors have produced (among other things) an important paper (ref 17) in which they explore the proteinase K-resistant domains of Sup35 amyloid isolated from a wide range of [PSI+] prion variants. This review explores the implications of this work. Their paper (ref. 17) is important because it examines the prion amyloid directly from cells, not amplified in vitro (a process which has not yet succeeded, in spite of the claims in ref. 84). And, as the authors point out, the previous work (using many structure-characterizing methods) on amyloid made de novo from E. coli – produced Sup35NM, has shown that the material is heterogeneous. At least the Kushnirov material is as close as one can hope to get to the real infectious material.
Critique:
1) The authors do not mention that the amyloid domains are folded along the long axis of the filaments. This is demonstrated by the diameter of the filaments (everyone’s filaments are around 5 or 6 nM) compared to the width of filaments if the beta sheet were unfolded (20 or so nm). It is also shown in classical X-ray diffraction of filaments as the ~10 Angstrom line, the average distance between adjacent sheets after the folding. The in-register parallel beta sheet can, in principle, fold in many different ways, even for the same location of the beta sheets.
2) The authors do not mention how prion amyloids template their structure. This may be the most interesting aspect of prions. Has anyone proposed a reasonable explanation of how this can occur?
3) Line 72: The authors point out that being “in-register” implies being “parallel”, so there is no reason to mention the “parallel” as it is redundant. That is true, but the authors leave out the possibility of “out-of-register parallel”. There are plenty of parallel beta sheets in globular proteins that have parallel beta sheets, but are not “in – register” because the strands of the sheet have different sequence. So there could be an out-of-register parallel beta sheet amyloid. It was shown that this is not the case in the yeast prion amyloids.
4) Lines 74-79 mention an “amyloid fold”. What is that? The authors write, “The in-register arrangement may not be optimal in terms of energy, for example, because similarly charged residues would be placed in close proximity.” The authors must know that the yeast prion domains are remarkably low on charged residues, and most mutants isolated by Cox and by Weissman that cannot form prions of Sup35 turned out to be changes from polar (Q) to charged residues. Then, “But such an arrangement is apparently the simplest that allows sequential reproduction of the fold.” How does that work? Is this the authors’ idea or is there a reference needed here? In fact, there are not references in section 2.1 “In-register arrangement” except for one about F-actin (not an amyloid) and one about the interaction of amyloid with chaperones and other proteins (not about “In-register arrangement”).
5) Line 130. “Some amyloids like Abeta sometimes can make both parallel and antiparallel fold [39].” The w.t. Abeta peptide has only been found to be parallel. Only peptides with mutant sequences can sometimes make some parallel filaments.
6) Line 38. Why are the neutral yeast prions so rare in wild type cells?
7) Line 39: Please give a few examples of reproducible advantages of yeast prions. This is important also for the authors’ argument that Hsp104, for example, should be called a “pro-prion” protein because it is necessary for prion propagation. If the prion really helps the cell, this would be reasonable. Tuite, in 1989, reported that [PSI+] made cells more heat-resistant and ethanol-resistant, but True and LIndquist did not find this in 5 pairs of strains. Advantages reported by True and Lindquist were not reproducible by Namy et al. using the same strains (Nat Cell Biol. 2008;10:1069 – 1075). By now, someone must have found cases where all agree that [PSI+], for example, is helpful to cells (other than in strains carrying nonsense mutations).
8) The discussion in Lines 248 to 274 of variants arising in certain mutants that cannot propagate in the wild type, does not mention the key point that in those mutants (hsp104T160M, upf, ssz1, zuo1, ssb1/2) the frequency of [PSI+] arising is >10-fold higher than in the wild type. The variants arising in the mutants that cannot propagate in the wild type are only part of the story. The usually studied variant types are also arising at a much higher frequency in those mutants.
Minor points:
Line 33: I suggest the authors also cite the work of Collinge’s group on actual Alzheimer’s disease infection in the patients receiving human growth hormone purified from cadavers and dying of CJD, Soto’s demonstration that amyloid of IAPP can transmit type II diabetes in animals, and the work of several groups that Parkinson’s disease is transmitted via the gut.
Line 174. Only one of the 19 fusions with aggregation properties were shown to be prions, [MOT3].
Lines 181-187. No references.
line 198, “[URE+]” should be [URE3].
Line 279: Magic angle spinning NMR and solid-state NMR are the same thing. In solid – state NMR, magic angle spinning is essential, and in solution NMR, there is no need for magic angle spinning.
Lines 436-449. The argument is somehow unclear to me, although I read it three times. Perhaps some re-wording would help, but I cannot suggest how, since I somehow still don’t get it.
In section 6, values of % soluble Sup35p in strong vs weak prion variant strains. I wonder if proper controls were done to show that the % aggregated was not overestimated by aggregation continuing in the extract. This particularly applies to the paragraph including Lines 573-578.
In summary, the authors have summarized some of the work on the structural bases of prion variation in yeast, emphasizing their own important work on this subject. However, inclusion of some of the other aspects outlined above would improve the usefulness of their review.
Author Response
We thank this Reviewer for his/her critical notes, which helped to improve the manuscript.
Review 1
of “Structural Bases of Prion Variation in Yeast” by Vitaly Kushnirov et al.
The authors have produced (among other things) an important paper (ref 17) in which they explore the proteinase K-resistant domains of Sup35 amyloid isolated from a wide range of [PSI+] prion variants. This review explores the implications of this work. Their paper (ref. 17) is important because it examines the prion amyloid directly from cells, not amplified in vitro (a process which has not yet succeeded, in spite of the claims in ref. 84). And, as the authors point out, the previous work (using many structure-characterizing methods) on amyloid made de novo from E. coli – produced Sup35NM, has shown that the material is heterogeneous. At least the Kushnirov material is as close as one can hope to get to the real infectious material.
Response: We thank the Reviewer for positive evaluation of our previous work. We tried here to present a general picture, rather than focusing on this work, but in frame of the topic of this review it was difficult to avoid frequently mentioning it.
Critique:
1) The authors do not mention that the amyloid domains are folded along the long axis of the filaments. This is demonstrated by the diameter of the filaments (everyone’s filaments are around 5 or 6 nM) compared to the width of filaments if the beta sheet were unfolded (20 or so nm). It is also shown in classical X-ray diffraction of filaments as the ~10 Angstrom line, the average distance between adjacent sheets after the folding. The in-register parallel beta sheet can, in principle, fold in many different ways, even for the same location of the beta sheets.
Answer: We agree with the reviewer. To make this point more clear, we have added an inset to the Figure 1, showing hypothetic protomer core folding (side view). As regards the exact values, we took the width of Sup35 amyloid core as 10 nm according to Baxa 2011 (PMID: 21219467, Ref. 115) and to Ghosh 2018 (PMID: 29846554, Ref. 102), while the unfolded beta sheet could be as long as 40 nm, considering that it is formed by the N domain of 122 residues with 4 nm of length per residue.
2) The authors do not mention how prion amyloids template their structure. This may be the most interesting aspect of prions. Has anyone proposed a reasonable explanation of how this can occur?
Answer: The molecular dynamics of how soluble disordered protein domain becomes a part of amyloid fold is unknown, or, at least, we are not aware of such works. We doubt that an experiment answering this question is currently technically possible. (If we understand the question in the same way as the Reviewer). But this question is not central to this review.
As we see the process, a disordered monomer molecule joins an amyloid fibril and acquires its fold simply because it is a lower energy state. It may be further speculated that, according to one of the works of the Lindquist lab (PMID: 17495929), some sites of a polypeptide chain have a higher propensity to start the process, and then the amyloid fold spreads from this site along a peptide chain to the borders of the disordered region. However, this description seems too speculative to be inserted into the text.
3) Line 72: The authors point out that being “in-register” implies being “parallel”, so there is no reason to mention the “parallel” as it is redundant. That is true, but the authors leave out the possibility of “out-of-register parallel”. There are plenty of parallel beta sheets in globular proteins that have parallel beta sheets, but are not “in – register” because the strands of the sheet have different sequence. So there could be an out-of-register parallel beta sheet amyloid. It was shown that this is not the case in the yeast prion amyloids.
Answer: We thank the reviewer for this suggestion. We have made an additional search and added three new references to Chapter 2.3 describing out-of-register parallel beta sheets in Abeta and amylin (IAPP).
Discussion: Regarding parallel beta sheets of globular proteins: we doubt that these could be used to make amyloid, as it was made from solenoid proteins with sequence repeats (Chapter 2.2.). Beta sheet can be formed by unsimilar beta strands taken from globular proteins, but there is no guarantee, and fairly doubtful, that these fragments would template their pro-amyloid fold from one fragment to another. The fragments would not automatically stick together to form amyloid.
4) Lines 74-79 mention an “amyloid fold”. What is that?
Answer: "Amyloid fold" refers to the core region, which forms intermolecular beta sheet.
The authors write, “The in-register arrangement may not be optimal in terms of energy, for example, because similarly charged residues would be placed in close proximity.” The authors must know that the yeast prion domains are remarkably low on charged residues, and most mutants isolated by Cox and by Weissman that cannot form prions of Sup35 turned out to be changes from polar (Q) to charged residues.
Answer: This is true, so we partly rewrote the text to discuss this:
“In terms of energy, the in-register arrangement is favorable for hydrophobic and aromatic residues, but is unfavorable for charged residues, which would experience repulsion when placed in close proximity. Charged residues are rare in yeast prions, but not that rare in human amyloids. For example, synuclein contains (in a 60 residue amyloid core) 11 charged residues, of which 6 are presumed to form intermolecular beta sheet (PDB: 2N0A, Ref. 30). Another consideration in favor of the frequent occurrence of the in-register arrangement could be that it is the simplest one that allows precise sequential reproduction of the fold.
Then, “But such an arrangement is apparently the simplest that allows sequential reproduction of the fold.” How does that work? Is this the authors’ idea or is there a reference needed here?
Answer: This is our idea, therefore no references. We think that this simplicity of the structure is equally or more important in defining the in-register arrangement than another reason, the self-affinity of some residues (e.g. hydrophobes).
In fact, there are not references in section 2.1 “In-register arrangement” except for one about F-actin (not an amyloid) and one about the interaction of amyloid with chaperones and other proteins (not about “In-register arrangement”).
Answer: We have added two references, for the first mentioning of parallel in-register (in Abeta) and for such structure in Sup35 (Refs 28 & 29).
5) Line 130. “Some amyloids like Abeta sometimes can make both parallel and antiparallel fold [39].” The w.t. Abeta peptide has only been found to be parallel. Only peptides with mutant sequences can sometimes make some parallel filaments.
Answer: This sentence and citation has been removed, but three new and more relevant citations of recent works were added. One of them describes antiparallel and parallel in- and out-of-register amyloids of wild type IAPP, while the other claims simultaneous presence in wt Abeta of antiparallel and parallel staggered structure(!) It is difficult to envision this, but the conclusions were based on NMR data.
6) Line 38. Why are the neutral yeast prions so rare in wild type cells?
Answer: We mean here that many, if not most of studied yeast prions are not obviously detrimental to their host cells. If you mean why prions are rare in yeast strains from nature, please see the next answer.
7) Line 39: Please give a few examples of reproducible advantages of yeast prions. This is important also for the authors’ argument that Hsp104, for example, should be called a “pro-prion” protein because it is necessary for prion propagation. If the prion really helps the cell, this would be reasonable. Tuite, in 1989, reported that [PSI+] made cells more heat-resistant and ethanol-resistant, but True and LIndquist did not find this in 5 pairs of strains. Advantages reported by True and Lindquist were not reproducible by Namy et al. using the same strains (Nat Cell Biol. 2008;10:1069 – 1075). By now, someone must have found cases where all agree that [PSI+], for example, is helpful to cells (other than in strains carrying nonsense mutations).
Answer: The question relates to the long-standing dispute between the schools of Reed Wickner and Sue Lindquist about whether yeast prions are a bug or a feature. So, we would like to describe our position on this.
Prions spoil the mechanisms related to them, which were optimized for the conditions most common to the yeast environment. Accordingly, they cannot be useful under such conditions and should have more or less negative effect. However, they can be important as a mechanism of adaptation to various adverse conditions, because they increase phenotypic variation. (An analogy coined by Reed Wickner: a soldier with a broken leg won't go to war and will have a higher chance to survive). Prions are likely to be present in a very small proportion of yeast population, but some of these cells can be more resistant to one unfavorable condition, and others to another and so on, and thus they have a better chance to survive such conditions and thus preserve local population. Surviving a bad time is no less important than efficient proliferation in a good time. Thus, prions represent a kind of bet hedging strategy. A specific advantage of prions is that they allow to return exactly to the standard phenotype after prion loss. Similar ideas in: PMID: 32896568, Ref.14.
The next question is whether prions can increase phenotypic diversity. If we disregard the mentioned works of True and Lindquist (which, we think, were not completely wrong), there are still some examples. [MOD5] improves resistance to some antifungals, [SWI] and [MOT3] alter the pattern of transcription, and so could be useful not just in adverse conditions. [GAR+] prion is involved in communication of yeast with bacteria (PMID: 25171409, 29861316). A good replacement for the True works would be the discovery of numerous presumably non-amyloid prions by Chakrabortee and Jarosz (PMID: 27693355, Ref. 73). In these cases, the phenotypic change was used to discover prions, and thus it was confirmed by this discovery and further analysis.
Thus, although we agree that prions are likely to be detrimental in a wide variety of conditions (although this effect might be quite subtle), there is sufficient evidence to think they are at least potentially useful under adverse conditions.
8) The discussion in Lines 248 to 274 of variants arising in certain mutants that cannot propagate in the wild type, does not mention the key point that in those mutants (hsp104T160M, upf, ssz1, zuo1, ssb1/2) the frequency of [PSI+] arising is >10-fold higher than in the wild type. The variants arising in the mutants that cannot propagate in the wild type are only part of the story. The usually studied variant types are also arising at a much higher frequency in those mutants.
Answer: We have added this consideration to the text and tried to avoid a clear statement on whether one or another concept of the considered effects is more correct, since this question is beyond the scope of the review. The added text:
“The frequency of prion appearance is usually several-fold higher in the mentioned strains with weakened or absent anti-prion proteins [83,88]. This indicates in favor of the anti-prion systems approach, and so possibly both concepts may be correct, depending on the particular “anti-prion” protein.”
Minor points:
Line 33: I suggest the authors also cite the work of Collinge’s group on actual Alzheimer’s disease infection in the patients receiving human growth hormone purified from cadavers and dying of CJD, Soto’s demonstration that amyloid of IAPP can transmit type II diabetes in animals, and the work of several groups that Parkinson’s disease is transmitted via the gut.
Answer: We have added the mentioned references.
Line 174. Only one of the 19 fusions with aggregation properties were shown to be prions, [MOT3].
Answer: Another is [NUP100] (Ref 71). Among others, some cannot be prions, because they were tested in an altered context (e.g. extracted from the middle of a protein to the N terminus as Gln3), as we discuss in Chapter 7.2. Others may be tried. No text was altered, since we see no conflict.
Lines 181-187. No references.
Answer: References were added (Refs 12, 62).
line 198, “[URE+]” should be [URE3].
Answer: Corrected
Line 279: Magic angle spinning NMR and solid-state NMR are the same thing. In solid – state NMR, magic angle spinning is essential, and in solution NMR, there is no need for magic angle spinning.
Answer: Thank you, we corrected the text.
Lines 436-449. The argument is somehow unclear to me, although I read it three times. Perhaps some re-wording would help, but I cannot suggest how, since I somehow still don’t get it.
Answer: The idea indeed was too complicated to be clear from the first attempt, and the conclusion was not certain. So we decided to remove this paragraph.
To clarify, we tried to find evidence in support of the ability of the region 124 to ~148 to form amyloid. The observation of PK resistant structure as such is an insufficient argument. However, Ohhashi et al observed PK-resistant cores at residues 81-148 or 62-144 in Sup35NM in vitro fibrils. These were the only cores within NM, and so they should include intermolecular beta sheet. However, there is still no guarantee that this beta sheet includes the region downstream of 124. We have also obtained better, but unpublished evidence: a prion based on one of the Sup35 scrambles of Ross and Wickner, where the only PK-resistant core includes the Sup35 region 115-132 plus three residues of the scrambled sequence. But we cannot review unpublished data.
In section 6, values of % soluble Sup35p in strong vs weak prion variant strains. I wonder if proper controls were done to show that the % aggregated was not overestimated by aggregation continuing in the extract. This particularly applies to the paragraph including Lines 573-578.
Answer: There are numerous problems with measuring the % of soluble Sup35, and we mention some of them in lines 522-529. We agree that the Sup35 aggregation could continue in lysate. One can stop aggregation with SDS, but this promotes Sup35 degradation. Then, one can delete Prb1 to prevent degradation and check that this does not affect the studied process. In our view, this discussion involves too many details, so we would prefer to leave this part as is.
In summary, the authors have summarized some of the work on the structural bases of prion variation in yeast, emphasizing their own important work on this subject. However, inclusion of some of the other aspects outlined above would improve the usefulness of their review.
Reviewer 2 Report
The review is devoted to the structural bases of the existence of different prion variants in yeast model object.
Introductory section contains enough information about prions for the understanding of further chapters of the review.
Second chapter describes possible amyloid conformation with the conclusion that parallel-in-register beta-structure is the one characteristic for PrP prion, and yeast amyloid prions.
The third chapter describes yeast prions as a model object for the study of prion properties in general which can help with the design of anti-prion drugs in case of human prion disease (and amyloid diseases). The similarity of life structure on a basic molecular level makes this concept real.
The fourth chapter is devoted to the phenomena of existence of prion variants in one of the most studied yeast prions called [PSI+]. Different factors affection the formation of [PSI+] variants are mentioned, and discussed.
The fifth chapter discusses possible structures of [PSI+] variants with a significant focus of the authors on the results of the identification of proteinase K-resistant cores in isolated amyloids of Sup35 corresponding to different variants.
Chapter six describes mechanisms involved in [PSI+] prion manifestation into phenotype, and chapter seven is devoted to identified [URE3], and [PIN+] prion variants.
The review is well-written, and covers all the topics on the subject of the review.
However, one minor improvement could be done in terms of the discussion of the role of anti-prion systems in prion variants generation. A possibility of such influence could be mentioned in the Discussion section. The authors discussed potential effects of anti-prion systems on generation of multiple [PSI+] variants with possible alternative interpretation of experiments involving T160M Hsp104 mutant. But they did not mention the existence of Btn2-Cur1 anti-prion system which represses a significant amount of [URE3] variants in wild type background. Also, a number of other observations on the influence of different factors on [PSI+] variants (including ribosome-associated Hsp70 Ssb1 and Ssb2, Upf1,2,3 nonsense-mediated mRNA decay factors, and Siw14 pyrophosphatase) mentioned by the authors in the review in combination with the data on above mentioned Btn2-Cur1 system rather points in the direction of the existence of the selection of prion variants upon the work done by anti-prion systems. For example, strong [URE3-1] variant is perfectly capable to propagate in btn2cur1 double knockout background, and the arguments applied for the studies done with T160M Hsp104 mutants can not be applied here. Some insights for the discussion could be found here: https://doi.org/10.1021/acs.biochem.7b01285
There is also a typo:
Line 307: “Cores 1 to 4, Figure 1B ” - There is no sections in Figure 1.
Overall, the review can be accepted for a publication considering that the authors will make a minor changes mentioned above.
Author Response
REVIEW 2
The review is devoted to the structural bases of the existence of different prion variants in yeast model object.
Introductory section contains enough information about prions for the understanding of further chapters of the review.
Second chapter describes possible amyloid conformation with the conclusion that parallel-in-register beta-structure is the one characteristic for PrP prion, and yeast amyloid prions.
The third chapter describes yeast prions as a model object for the study of prion properties in general which can help with the design of anti-prion drugs in case of human prion disease (and amyloid diseases). The similarity of life structure on a basic molecular level makes this concept real.
The fourth chapter is devoted to the phenomena of existence of prion variants in one of the most studied yeast prions called [PSI+]. Different factors affection the formation of [PSI+] variants are mentioned, and discussed.
The fifth chapter discusses possible structures of [PSI+] variants with a significant focus of the authors on the results of the identification of proteinase K-resistant cores in isolated amyloids of Sup35 corresponding to different variants.
Chapter six describes mechanisms involved in [PSI+] prion manifestation into phenotype, and chapter seven is devoted to identified [URE3], and [PIN+] prion variants.
The review is well-written, and covers all the topics on the subject of the review.
However, one minor improvement could be done in terms of the discussion of the role of anti-prion systems in prion variants generation. A possibility of such influence could be mentioned in the Discussion section. The authors discussed potential effects of anti-prion systems on generation of multiple [PSI+] variants with possible alternative interpretation of experiments involving T160M Hsp104 mutant. But they did not mention the existence of Btn2-Cur1 anti-prion system which represses a significant amount of [URE3] variants in wild type background. Also, a number of other observations on the influence of different factors on [PSI+] variants (including ribosome-associated Hsp70 Ssb1 and Ssb2, Upf1,2,3 nonsense-mediated mRNA decay factors, and Siw14 pyrophosphatase) mentioned by the authors in the review in combination with the data on above mentioned Btn2-Cur1 system rather points in the direction of the existence of the selection of prion variants upon the work done by anti-prion systems. For example, strong [URE3-1] variant is perfectly capable to propagate in btn2cur1 double knockout background, and the arguments applied for the studies done with T160M Hsp104 mutants can not be applied here. Some insights for the discussion could be found here: https://doi.org/10.1021/acs.biochem.7b01285
Answer: Thank you for this helpful suggestion. We introduced the offered information and added this and another reference (Refs 86,87) to the text.